# Kinetics of Isothermal Dumbbell Exponential Amplification: Effects of Mix Composition on LAMP and Its Derivatives

**DOI:** 10.3390/bios12050346

**Published:** 2022-05-18

**Authors:** Maud Savonnet, Mathilde Aubret, Patricia Laurent, Yoann Roupioz, Myriam Cubizolles, Arnaud Buhot

**Affiliations:** 1Univ. Grenoble Alpes, CEA, CNRS, IRIG-SyMMES, 38000 Grenoble, France; maud.savonnet@grenoble-inp.org (M.S.); mathilde.aubret@cea.fr (M.A.); yoann.roupioz@cea.fr (Y.R.); 2Microfluidic Systems and Bioengineering Lab, Technologies for Healthcare and Biology Department, Univ. Grenoble Alpes, CEA, LETI, 38000 Grenoble, France; patricia.laurent@cea.fr

**Keywords:** loop mediated isothermal amplification (LAMP), isothermal dumbbell exponential amplification (IDEA), kinetic model, nucleic acids amplification, logistic function

## Abstract

Loop-mediated isothermal amplification (LAMP) is an exponential amplification method of DNA strands that is more and more used for its high performances. Thanks to its high sensitivity and selectivity, LAMP found numerous applications from the detection of pathogens or viruses through their genome amplification to its incorporation as an amplification strategy in protein or miRNA biomarker quantification. The LAMP method is composed of two stages: the first one consists in the transformation of the DNA strands into dumbbell structures formed of two stems and loops thanks to four primers; then, in the second stage, only two primers are required to amplify the dumbbells exponentially in numerous hairpins of increasing lengths. In this paper, we propose a theoretical framework to analyze the kinetics of the second stage of LAMP, the isothermal dumbbell exponential amplification (IDEA) as function of the physico-chemical parameters of the amplification reaction. Dedicated experiments validate the models. We believe these results may help the optimization of LAMP performances by reducing the number of experiments necessary to find the best parameters.

## 1. Introduction

The polymer chain reaction (PCR) amplification method developed by Mullis in 1986 [1,2] allows for exponential multiplication of a template DNA sequence thanks to an enzymatic reaction and two specific primers recognizing the endpoints of the template. Thanks to several temperature cycles, this enzymatic reaction enables to produce a large amount of DNA copies starting from a very small quantity of initial sequences in a complex sample (blood, saliva, urine, food matrix, etc.). However, in addition to be sensitive to various inhibitors, this biomolecular reaction requires cycles at different temperatures, which remains an important drawback in particular for its integration in a portable device. Therefore, several isothermal amplification methods have been developed in the last few decades [3]. Among them, the famous loop-mediated isothermal amplification (LAMP) has been proposed by Notomi et al. in the early 2000s [4]. In the on-going COVID-19 pandemic, the LAMP method has been recognized as an efficient and rapid diagnostic technique for viral genome detection and a strong alternative to the standard PCR [5,6,7,8,9]. Unlike PCR, LAMP does not require any temperature cycle but is based on the use of four different primers. The LAMP experiments are generally decomposed in two stages (see Appendix A for an illustration of the first steps). The first one transforms the target strands with the help of four or six primers into a dumbbell structure composed of two stems and two loops. Two outer primers (generally called B3 and F3) help to produce the dumbbell structure with the two remaining primers, backward and forward inner primers (BIP and FIP, respectively). The primers B3 and F3 are necessary to create and release the dumbbell structure from the target strand. The second stage consists in the isothermal dumbbell exponential amplification (IDEA). The two BIP and FIP primers allow for an exponential amplification of the number of amplicons or hairpins structures with an increasing stem length thanks to an enzyme (generally Bst DNA polymerase) at a fixed temperature (in the range 60 to 70 ${}^{\circ }$C) [4,10]. The complete LAMP reaction requires four primers but two other primers called loop primers are sometimes added to accelerate the reaction. The LAMP method allows for an isothermal, rapid, sensitive, specific and robust amplification of nucleic acid sequences [5,11] with limited sensitivity to inhibitors [12] and with possibility of integration for Point-of-Care devices [13,14,15]. LAMP seems to be the most promising alternative to PCR as the most specific and sensitive method of all isothermal amplification techniques [16]. LAMP was initially used for the detection of viruses and pathogens of interest in a sample [17,18,19,20]. More recently, new applications appeared for the detection of protein biomarkers [21,22], miRNAs [23,24,25,26,27,28], single base mutations [29,30,31] or methylation content [32]. In such applications, the traditional LAMP method is generally modified to take advantage of its second stage providing the exponential amplification signal [33,34,35,36,37,38].

Thanks to its high sensitivity, the LAMP reaction allows for the detection of as low as a single DNA copy from a complex sample in only few tens of minutes [39,40]. However, optimizing all the parameters influencing the LAMP reaction in order to get the fastest and most efficient amplification is a long and laborious process [41]. Indeed, many biochemical parameters influence the overall amplification cascade: the concentration of the enzyme, the primers, the nucleotides or other chemical compounds like salts or magnesium sulfate. Physical parameters such as the temperature can also affect the reaction. For each DNA target amplification or diagnostic application, a new set of experiments must be performed to find the best parameters since no systematic rules can be applied for the exponential growth in LAMP reaction. However, building a mathematical model based on the physico-chemical parameters, which could simulate the kinetic reaction, would certainly help to avoid many repetitive optimization experiments. While for the PCR reaction, the different temperature cycles are rendering the exponential amplification straightforward with a (nearly) doubling of DNA copies or strands per cycle [42,43,44], the LAMP reaction is a complex isothermal reaction producing hairpins of various lengths. Thus, the full reaction kinetics should be analyzed. Only recently, this topic has attracted the interest of some scientists [45,46,47]. Most of the approaches were empirical and mainly based on (generalized) logistic functions, also known as the Richards curve, whose formula is $S\left(t\right)=S_{max}/\left[1+\exp\left(k\left(t_{50}-t\right)\right)\right]$. The real-time signal (generally fluorescence) S(t) is related to the amount of amplicons (hairpin strands produced during the amplification reaction). $S_{max}$ is the saturation signal at late times while *k* is the exponential growth rate and $t_{50}$ is the time at which 50% of the maximum signal is achieved. Both parameters *k* and $t_{50}$ are generally fitted to the experimental results [46].

The aim of this paper is to provide a general framework to assess the viability of such a model logistic function to analyze LAMP experiments. We focused on the second stage of the LAMP method called IDEA for isothermal dumbbell exponential amplification, neglecting the steps of the first stage to produce the dumbbells. We have developed a theoretical model from the basic principles of the reaction kinetics in order to interpret the experimental signal growth in terms of physico-chemical parameters (principally the concentration of dumbbells, primers and nucleotides). We observed that different regimes may occur depending on those concentrations. Our simplest model based on primer hybridization limiting reaction rate predicts that the increase of the amplicons (or hairpins) concentration follows a logistic function. This allows us to predict the dependence of *k* and $t_{50}$ on the concentration of dumbbell and primers. We performed various experiments in order to validate some of our predictions.

## 2. Materials and Methods

### 2.1. Oligonucleotides: Dumbbells and Primers

As mentioned in the introduction, we focus on the second stage of the LAMP method, the IDEA. By initiating the LAMP reaction with dumbbells, we are able to bypass completely the first stage requiring four primers. Thus, only the two primers Backward Inner Primer (BIP) and Forward Inner Primer (FIP) are required for IDEA. The dumbbell probes are composed of single stranded oligonucleotides with a double stem loop structure. The short dumbbell (SD) structure presents the minimal length (134 nucleic acid bases) compatible with their corresponding primers. Its sequence is defined as follows 5${}^{\prime }$-F1c-F2-F1-B1c-B2c-B1-3${}^{\prime }$. F1c is the complementary strand of F1, B1c the complementary strand of B1 and B2c the complementary strand of B2 (see Table 1 and Figure 1 for details of the sequences).

To study the effect of the dumbbell length, we considered two other dumbbells incorporating extensions. In several portions of this SD sequence, other sequences have been introduced enabling the amplification with the same couple of primers, BIP and FIP. The middle dumbbell (MD) (153 bases) incorporates the extension sequence Ext1a (16 bases) in the left loop between F2 and F1. It also contains the sequence named Ext2 (the 3 bases TGA) in the middle between F1 and B1c. The long dumbbell (LD) (190 bases) presents several extension sequences: Ext1b (23 bases) in the left loop between F2 and F1, Ext2 (3 bases TGA) in the middle between F1 and B1c, and Ext3 (30 bases) in the right loop between B1c and B2c. All the oligonucleotides were purchased from Eurogentec (France).

### 2.2. Isothermal Dumbbell Exponential Amplification (IDEA)

The LAMP reactions starting from a dumbbell are called IDEA. Every reaction was carried out within 20 µL working volume composed of 18 µL of LAMP mix solution and 2 µL of the dumbbell solution at a fixed concentration. The LAMP mix solution was composed of several reagents with final concentrations as follows except otherwise specified: FIP and BIP at 2.4 µM, 1X Isothermal amplification Buffer (NEB, France), 1 mM MgSO${}_{4}$ (NEB, France), 0.8 M Betain (Sigma Aldrich, France), 1.4 mM of deoxyribonucleotide tri-phosphates (dNTPs) solution (Sigma Aldrich, France), 0.4 U/mL of Bst 2.0 DNA polymerase (NEB, France), and 1X of Eva Green Fluorescent DNA intercalating dye (Jena Bioscience, Germany). Finally, the solutions were placed in the QuantStudio 3 Real-Time PCR System (ThermoFisher) and heated at 65${}^{\circ }$C for one hour. The fluorescence *F* of the solution was recorded every 30 s. Furthermore, for each sample, duplicate amplifications were systematically performed. The normalized fluorescence was defined as $\Delta F=\left(F-F_{min}\right)/\left(F_{max}-F_{min}\right)$ where $F_{min}$ and $F_{max}$ are, respectively, the minimal and maximal fluorescence values of the raw data (see Appendix A for an illustration).

## 3. Theoretical Model for IDEA Kinetics of LAMP Experiments

### 3.1. Evolution Rules for IDEA

First, we consider that Hp1b hairpins are created from the dumbbells (see Figure 1). In the name Hp1b, Hp stands for Hairpin, 1 for a stem containing one dumbbell sequence and b for a B2c loop compatible with a BIP sequence. Such Hp1b hairpin formation requires only the help of the enzyme (Bst DNA polymerase working at 65 ${}^{\circ }$C). From this hairpin Hp1b, we may expect that two new hairpins Hp1f and Hp2f are produced by the enzymes from the use of a single BIP primer (see Figure 2):(1)$$\mathrm{Hp}1b+\mathrm{BIP}\underset{k_{B}}{\to }\mathrm{Hp}1f+\mathrm{Hp}2f$$
where $k_{B}$ is the kinetic rate constant of the reaction. Such duplication of hairpins is a key step in the exponential amplification of the number of hairpins or amplicons. In the same way, two hairpins Hp1b and Hp2b are created from Hp1f (oligonucleotide sequence containing a F2c loop compatible with FIP sequence), with the help of a FIP primer and Bst enzymes. In both cases, Hp2f and Hp2b are hairpins with 2 dumbbell sequences in the stem (f stands for F2c and b for B2c in the loop).

We can generalize the first duplication reaction (Figure 2) to the creation of hairpins with various stem lengths (see an illustration in Steps 7 and 8 of Appendix A). Each hairpin Hp$2^{n}$x with n≥1 and x = f or b is duplicated in two hairpins Hp$2^{\left(n+1\right)}$y and Hp$2^{n}$y (where y = f if x = b and vice-versa) with the help of one single compatible primer sequence FIP or BIP. Let us consider the example of the hairpin Hp2f, the binding of the primer FIP followed by enzyme prolongation of the 3’ ends leads to two different hairpins: Hp2b and Hp4b. Then, for n=2, the hairpin Hp$2^{2}$f (=Hp4f) is duplicated in Hp$2^{\left(2+1\right)}$b (=Hp8b) and Hp$2^{2}$b (=Hp4b). If the initial hairpin was compatible to BIP and contained a B2c loop, then the newly created hairpins are compatible to FIP and vice-versa. The respective kinetic rates $k_{F}\left(n\right)$ and $k_{B}\left(n\right)$ may depend on the length of the hairpin (represented by *n*) and on the primers, BIP or FIP, considered. One of the created hairpin has the same stem length as the initial one and the other presents a length (nearly) two times longer than the initial one. For n≥1, the length of the stem of Hp$2^{n}x$ contains $2^{n}$ sequences of the dumbbell with the F1-F1c stem for hairpins with F2c loop (see Hp2F in Figure 2) or B1-B1c for hairpins with B2c loop. The case of n=0 is particular since only half of the dumbbell and its complementary are present in the stem (see Hp1f and Hp1b in Figure 2). The doubling of the hairpin number and the exponential increase of the length of the hairpins both explain the fast exponential amplification of LAMP method.

Figure 2 depicts the duplication reaction for SD without extension of sequences. In the case of MD and LD where extensions are present, the first steps are slightly different (See Appendix A the structures of the hairpins Hp1f, Hp2f and Hp1b for the dumbbell LD). From Figure 1, it is straightforward that the complementary strands of extensions Ext1 and Ext2 are directly incorporated inside Hp1b. Then, the complementary strand Ext3c of extension Ext3 is incorporated by the enzyme after the binding of the primer BIP into the hairpin Hp1b so that the hairpin Hp2f presents the three extensions (or their complementary strands) in the 1st and 2nd dumbbells depicted in Appendix A.

From those duplication rules of hairpins, we have developed a mathematical model of the IDEA kinetics based on simple arguments for the limiting kinetic rates. In the following sections, we will discuss the different possible regimes of the amplification reaction and deduce the dependence of these rates on relevant parameters.

### 3.2. Primer Limiting Kinetic Rates

In the first model, the kinetic rates are supposed to be limited by the hybridization of the primer to the hairpin loops. Such hybridization rate is known to be constant $k_{hyb}≃0.1-10\;\mu$M${}^{-1}$· s${}^{-1}$ and almost independent of the sequence and length of the duplex formed [48]. Thus, for simplicity, we remove the length and primer dependence and only consider $k_{F}\left(n\right)=k_{B}\left(n\right)=k_{hyb}$. Finally, with P standing for BIP or FIP, the kinetics rules are set by:(2)$$\mathrm{Hp}2^{n}+P\underset{k_{hyb}}{\to }\mathrm{Hp}2^{n+1}+\mathrm{Hp}2^{n}.$$

Let c(n,t) be the concentration of Hp$2^{n}$ hairpins with *n* standing for the length of the stem containing $2^{n}$ dumbbell sequences and $c_{p}\left(t\right)$ the concentration of primers as function of time *t*. If we neglect the transformation time of dumbbells into Hp1b hairpins, the initial conditions are $c\left(0,0\right)=c_{D}$ where $c_{D}$ is the initial concentration of dumbbells, c(n≥1,t)=0 and $c_{p}\left(0\right)=c_{P}=c_{FIP}+c_{BIP}$ with $c_{P}$ the initial concentration of primers and $c_{FIP}$ and $c_{BIP}$ the initial concentrations of FIP and BIP primers, respectively. With $l_{D}$ the length of the dumbbell, the length of an hairpin Hp1 is $l_{Hp1}≃3l_{D}/2$ since half of the dumbbell has been elongated as visible in Figure 2. This approximation for Hp1f and Hp1b becomes exact when considering the average length of the hairpins Hp1 since $l_{Hp1b}+l_{Hp1f}=3l_{D}$. Similarly, the length of an hairpin Hp$2^{n}$ is $\left(2^{n}+1/2\right)l_{D}$. Thus, the concentration $c_{Hp}$ and the total length $l_{Hp}$ of hairpins in solution are defined by:(3)$$c_{Hp}\left(t\right)=\sum_{n=0}^{\infty }c\left(n,t\right)\;\mathrm{and}\;l_{Hp}\left(t\right)=l_{D}\sum_{n=0}^{\infty }2^{n}c\left(n,t\right)+l_{D}\times c_{Hp}\left(t\right)/2$$
where the total concentration $c_{Hp}$ of hairpins is the sum of the concentrations of hairpins with fixed stem length represented by *n*. The total length $l_{Hp}$ is decomposed in two terms: the first one mainly represent the stem length (with $2^{n}l_{D}$ the stem length of an hairpin Hp$2^{n}$) while the loop part is constant for all hairpins leading to the product $l_{D}\times c_{Hp}\left(t\right)$. To be completely rigorous, the second term incorporates a small part of the stem close to the loop (F1-F1c or B1-B1c depending on the hairpin BIP or FIP considered). From Equation (Equation 2), the kinetic rules translate to the following set of differential equations:(4)$$\frac{d}{dt}c\left(0,t\right)=0\;\mathrm{and}\;\frac{d}{dt}c\left(n,t\right)=k_{hyb}\;c_{p}\left(t\right)\times c\left(n-1,t\right)\;\mathrm{for}\;n\geq 1$$
leading to single order differential equations for $c_{Hp}$ and $l_{Hp}$ (see the Appendix A): (5)$$\begin{array}{ccc} \frac{d}{dt}c_{Hp}\left(t\right) & = & k_{hyb}c_{p}\left(t\right)c_{Hp}\left(t\right) \end{array}$$(6)$$\begin{array}{ccc} \frac{d}{dt}l_{Hp}\left(t\right) & = & k_{hyb}c_{p}\left(t\right)\left(2l_{Hp}\left(t\right)-l_{D}c_{Hp}\left(t\right)/2\right) \end{array}$$
Indeed, if we assume that the rate constant $k_{hyb}$ does not depend on the hairpin length, the differential equations for $c_{Hp}\left(t\right)$ and $l_{Hp}\left(t\right)$ can be combined to lead:(7)$$l_{Hp}\left(t\right)=l_{D}\left(\frac{c_{Hp}^{2}\left(t\right)}{c_{D}}+\frac{c_{Hp}\left(t\right)}{2}\right)$$
with $c_{D}$ the initial concentration of dumbbells. From this equation (see the calculation in the Appendix A), the total length of hairpins $l_{Hp}$ increases quadratically with the concentration of hairpins $c_{Hp}$ illustrating the increase of the average length of hairpins.

#### 3.2.1. Constant Primer Concentration

Assuming a large concentration of primers, we may consider at least for short times that this concentration remains constant: $c_{p}\left(t\right)=c_{P}$. Then, the equation for the concentration of hairpins follows from the integration of Equation (Equation 5):(8)$$c_{Hp}\left(t\right)=c_{D}\exp\left(k_{hyb}c_{P}t\right).$$

As expected, both the concentration and length of hairpins are exponentially increasing with a kinetic rate two times larger for the length than for the concentration of hairpins. In both cases, the exponential rate *k* is proportional to $k_{hyb}$, which corresponds to the hybridization rate of the primers, and $c_{P}$ their concentration. These predictions can be experimentally tested by varying the primer concentrations (See Section 4).

#### 3.2.2. Saturation Due to Finite Primer Concentration

Since the number of initial primers is finite, at long timescales, the hairpin concentration may saturate due to a lack of primers. A conservation law relates both concentrations of hairpins and primers. Indeed, for each new hairpin produced, a primer is disappearing. Thus, $c_{Hp}\left(t\right)+c_{p}\left(t\right)=c_{D}+c_{P}=c_{tot}$ and the differential equation becomes:(9)$$\frac{d}{dt}c_{Hp}\left(t\right)=k_{hyb}\left(c_{tot}-c_{Hp}\left(t\right)\right)c_{Hp}\left(t\right).$$
With these assumptions, the concentration of hairpins $c_{Hp}\left(t\right)$ follows a logistic function (see complete calculation in the Appendix A) as generally fitted for LAMP experiments [46]:(10)$$c_{Hp}\left(t\right)=\frac{c_{tot}c_{D}\exp\left(k_{hyb}c_{tot}t\right)}{c_{P}+c_{D}\exp\left(k_{hyb}c_{tot}t\right)}=\frac{c_{tot}}{1+c_{P}\exp\left(-k_{hyb}c_{tot}t\right)/c_{D}}$$
where $c_{tot}=c_{D}+c_{P}$ is the sum of the dumbbell and primer initial concentrations. As can be seen from this equation, at the initial time, $c_{Hp}\left(0\right)=c_{D}$ and the number of hairpins saturates to $c_{Hp}\left(\infty \right)=c_{tot}$ with an exponential increase at short and intermediate times, such as in Equation (Equation 8). Indeed, $c_{tot}=c_{D}+c_{P}≃c_{P}$ since the primer concentration (usually in the micromolar range) largely exceeds the dumbbell concentrations in most LAMP experiments. Thus, the concentration of hairpins follows a logistic function with a growth rate $k=k_{hyb}c_{tot}≃k_{hyb}c_{P}$ and a time at half saturation $t_{50}=\ln\left(c_{P}/c_{D}\right)/k_{hyb}c_{P}$ scaling logarithmically with the dumbbell concentration as generally observed in LAMP experiments. The total length of hairpins $l_{Hp}\left(t\right)$ may be expressed as function of $c_{Hp}\left(t\right)$ following Equation (Equation 7) and is not following explicitly a logistic function due to its quadratic dependence upon $c_{Hp}\left(t\right)$.

In this first model where the limiting parameters is the primer concentrations, we assumed that the concentration of dNTPs is sufficient throughout the amplification. This assumption may easily be checked from the amount of hairpins produced during an amplification.

### 3.3. Saturation Due to Finite dNTPs Concentration

Since the amount of dNTPs is also finite, at long timescales, the hairpin concentration may reach a plateau due to a lack of dNTPs before the primer concentrations are limiting. After considering the hybridization of primers as the limiting step, we thus need to consider in this section that the enzymatic rate may become the limiting step due to the rapid decrease in dNTP concentration. Considering the enzyme reaction as the limiting kinetic rate, the Michaelis–Menten reaction constant should be considered:(11)$$k_{MM}=k_{E}c_{n}\left(t\right)/\left(c_{n}\left(t\right)+K_{D}\right)$$
with $c_{n}\left(t\right)$ the concentration of dNTPs, $K_{D}$ a dissociation constant generally measured around few tens of micromolars and $k_{E}$ the enzyme reaction rate in the range of 10–1000 s${}^{-1}$[49,50]. There are two different regimes for the Michaelis–Menten rate: first, when $c_{n}\left(t\right)\gg K_{D}$ at the initial stage of the amplification, the enzymatic rate is constant and equals to $k_{E}$. At longer times, the dNTP concentration decreases and becomes limiting leading to a concentration dependent rate $k_{MM}≃k_{E}c_{n}\left(t\right)/K_{D}$ for $c_{n}\left(t\right)<K_{D}$. Finally, a saturation of the hairpin concentration occurs when the dNTP concentration vanishes. Since $k_{hyb}c_{P}<k_{E}$ for primer concentrations in the micromolar range as usually observed in experiments, the first model is relevant at short times of amplification. Then, a cross-over from the primer-limited to the dNTP-limited regimes could be observed at time $t_{co}$ when the kinetic rate of primer hybridization $k_{hyb}c_{P}$ equals the Michaelis–Menten kinetic rate $k_{MM}$. This cross-over occurs for dNTP concentrations $c_{n}\left(t_{co}\right)=k_{hyb}c_{P}K_{D}/k_{E}$ setting the cross-over time $t_{co}$.

In this section, we assume that the concentration of primers is sufficient and constant throughout the amplification. This assumption may easily be checked by determining the amount of hairpins produced during an amplification and the number of dNTPs incorporated in those hairpins. Contrarily to the primer concentration, the dNTP concentration is related to the total length of hairpins $l_{Hp}\left(t\right)$ instead of their concentration $c_{Hp}\left(t\right)$. Thus, with this assumption, we need the relationship between $l_{Hp}\left(t\right)$ and $c_{n}\left(t\right)$. The concentration of dNTPs is directly related to the length of hairpins produced because a number of dNTPs proportional to the length of the hairpin produced disappears for each new hairpin produced: $c_{n}\left(t\right)=c_{n0}+l_{Hp}\left(0\right)-l_{Hp}\left(t\right)$ with $c_{n0}$ the initial concentration of dNTPs if we neglect the length of the primers incorporated into the hairpins. We can see this equation as a simplification of the conservation law of the total number of nucleotides in the solution: $c_{n}\left(t\right)+l_{Hp}\left(t\right)+l_{P}c_{p}\left(t\right)=c_{n0}+l_{Hp0}+l_{P}c_{P}$ with $l_{P}$ the length of the primers. Although a set of coupled differential equations combining the length of hairpins $l_{Hp}\left(t\right)$ and the concentrations $c_{p}\left(t\right)$ and $c_{n}\left(t\right)$ of primers and dNTPs should be considered, for simplicity, we decided to consider a constant primer concentration.

Replacing the primer limiting kinetic rate $k_{hyb}c_{p}\left(t\right)$ by the Michaelis–Menten equation $k_{MM}$ (Equation (Equation 11)) with the dNTPs concentration $c_{n}\left(t\right)$ as function of $l_{Hp}\left(t\right)$ inside Equation (6), we obtain the following equation:(12)$$\frac{d}{dt}l_{Hp}\left(t\right)=\frac{2k_{E}\left(l_{tot}-l_{Hp}\left(t\right)\right)l_{Hp}\left(t\right)}{l_{tot}-l_{Hp}\left(t\right)+K_{D}}$$
where $l_{tot}=c_{n0}+l_{Hp}\left(0\right)$. For simplicity, we neglect $l_{D}\times c_{Hp}\left(t\right)$ compared to $l_{Hp}\left(t\right)$ since we have seen that the growth of $l_{Hp}\left(t\right)$ is faster than $c_{Hp}\left(t\right)$. An integration on the timescale $\left[t_{co},t\right]$ leads to (see the Appendix A):(13)$$\exp\left(2k_{E}\left(t-t_{co}\right)\right)={\left(\frac{l_{Hp}\left(t\right)}{l_{Hp}\left(t_{co}\right)}\right)}^{\frac{l_{tot}+K_{D}}{l_{tot}}}{\left(\frac{l_{tot}-l_{Hp}\left(t_{co}\right)}{l_{tot}-l_{Hp}\left(t\right)}\right)}^{\frac{K_{D}}{l_{tot}}}.$$
This equation differs from a logistic function due to the two terms on the right side. Generally, $l_{tot}≃c_{n0}$, which is in the milimolar range, while $K_{D}$ is in the micromolar range. Thus, $l_{tot}\gg K_{D}$ allowing simplifications. Furthermore, as can be seen from the solution, $l_{Hp}\left(\infty \right)$ saturates to $l_{tot}$ with an exponential increase at times above $t_{co}$:(14)$$l_{Hp}\left(t\right)≃l_{Hp}\left(t_{co}\right)\exp\left(2k_{E}\left(t-t_{co}\right)\right).$$
The hairpin length increases exponentially with a rate $2k_{E}$ before saturation due to a lack of dNTPs. This rate depends on the enzyme and no more on the concentration of primers.

### 3.4. Guidelines for Practitioners

In the previous sections, we have developed a theoretical understanding of the second stage of LAMP, named IDEA. In this section, we aim to (i) discuss the main assumptions and approximations of the model, (ii) highlight the key parameters to improve the amplification and (iii) describe the experimental techniques to analyze and measure the hairpin concentrations and lengths.

(i)Main assumptions and approximations of the model and their consequences:

**Assumption** **A1.**
*We simplified the complex LAMP reaction by the succession of duplication reactions: the formation of two hairpins from a single one by the help of a primer (see Equation (Equation 2)). This duplication reaction is in fact the combination of several reactions starting from a primer binding to the hairpin loop followed by three enzyme elongations to form the two hairpins (Figure 2). This assumption is based on the consideration that either the primer binding or the enzyme elongation was the limiting reaction in the duplication reaction. Thus, it is directly related to the prediction of the primer-limited or dNTP-limited regimes.*


**Assumption** **A2.**
*We considered only hairpins making the hypothesis that the LAMP are forming the most stable structures. However, during enzyme elongation the folding of structures with multiple loops could also be present [47]. In this case, multiple primers could hybridize with the loops modifying the Equation (Equation 2).*


Approximations: For simplification of the model, we also performed multiple approximations. The enzymes were expected to remain active throughout the amplification. Like in PCR model, we could consider that the fraction of active enzymes is decaying exponentially with time [43]. We neglected the difference between BIP and FIP primers binding to the hairpins. Their initial concentrations is generally equal in LAMP experiments, however, their hybridization rate $k_{hyb}$ to their respective loop could slightly differ. In this case, their decay would be different. Similarly, for the dNTPs, we did not specified the relative concentrations of the four bases. However, depending on the GC content of the dumbbell, the decay rate of G and C could differ from the one of A and T. Such approximations are particularly important to analyze the saturation of the primer-limited or dNTP-limited regimes, respectively.

(ii)Key parameters for LAMP optimization:

*Primer concentration.*The primer concentration is involved in the growth rate *k* of the concentration and length of hairpins. Thus, a slight modification of their concentration would affect strongly the LAMP. While a higher concentration would accelerate the amplification, it is important to mention that the drawback is a potential binding between primers that would lead to primer-dimers and non-specific amplification [34,51].

*dNTP concentration.* As predicted by the model, a dNTP-limited regime appeared after the primer-limited regime at a crossover time $t_{co}$. Due to their large concentration in experiments (in the milimolar range), it should not affect the positive signal of amplification for end-point measurements (either by colorimetry, fluorescence, pH level or turbidity) [34]. For real-time measurements, the time-to-positive (time when the experimental signal is above a threshold at least three times the experimental noise) should occur in the first primer-limited regime.

*Enzyme activity and concentration.* Due to the Assumption 1, the primer-limited regime is independent from the enzyme. Only the second dNTP-limited regime is dependent on the enzyme elongation rate. Such assumption was required to obtain a simple solution of the first regime. It should be reconsidered in order to take into account explicitly the influence of the enzyme activity and concentration in this regime.

*Temperature.* Generally, LAMP is performed in the range 60 to 70 ${}^{\circ }$C. Even if we did not explicitly mentioned the temperature, its influence on the various affinity constants is obvious. For example, the temperature affects the binding of primers to loops through $k_{hyb}$ as well as the enzyme rate of elongation $k_{E}$.

*Other components.* The other components of the LAMP mix have not been taken into account explicitly but they play an indirect role. For example, salts certainly affect the hybridization (thermodynamics and kinetics) of the hairpin stems as well as of the primers to the loops.

(iii)Experimental techniques to analyze LAMP:

Three main categories of experimental techniques exist to analyze the exponential growth of LAMP and they are measuring different parameters:1/The most standard technique is the use of intercalating dyes such as EvaGreen or SYBR Green. From the change of fluorescence due to the binding of the dye to the stem of hairpins, it is possible to measure the amount of stem length in the solution. By neglecting the loop parts (second term in the expression of $l_{H}p$ in Equation (Equation 3)), we may consider that the measured fluorescence is proportional to the total length of hairpins $l_{Hp}$. We have considered this technique to validate our model later on.2/The detection of amplification by release of quenching (DARQ) is based on the use of a complementary strand to one of the primer and a couple of quencher-fluorophore attached on each strand [34,52]. Before the amplification, the fluorophore is quenched and no (or little) fluorescence is observed. During LAMP amplification, the complementary strand of the primer is released when the later is incorporated into a hairpin. Thus, in this case, the fluorescence is proportional to the concentration $c_{Hp}$ of hairpins formed during the amplification. Similar techniques have been developed [34].3/Finally, by gel electrophoresis, the amount of hairpins of different stem lengths c(n,t) could be analyzed. This method is generally considered at the end of the amplification where different bands are observed corresponding to the different stem lengths present in the solution. In the dNTP-limited regime, the lack of dNTP could lead to a smearing of the band due to interrupted elongation of hairpin stems.

### 3.5. Fluorescence Detection with Intercalating Dyes

As discussed in the previous section, a standard experimental technique to monitor the total length of hairpin consists in adding a fluorescent dye such as EvaGreen or SYBR green [4,34,53]. A recent model for PCR amplification was explicitly considering this dye-based detection [54]. However, due to the finite concentration of fluorescent dyes and its dissociation constant $K_{dye}$ towards double stranded DNA (dsDNA) [55], the observed fluorescence was not directly proportional to the length of the dsDNA in solution but may saturate due to the incorporation of most of the dyes inside the dsDNA strands. The same saturation may occur in LAMP experiments where the dyes are intercalating in the stem of the hairpins. Let us define the minimal distance $l_{0}$ between two dyes inside a stem of hairpin. This parameter $l_{0}$ allows taking into account the stoichiometry of the dyes towards a stem as function of its length. The concentration of accessible sites is increasing with the hairpin length with the ratio $l_{Hp}\left(t\right)/l_{0}$. Due to the affinity $1/K_{dye}$ of the dyes towards dsDNA, the fraction α of dyes bound to hairpins is given by the law of mass action:(15)$$\left(1-\alpha \right)\left(l_{Hp}\left(t\right)/l_{0}-c_{dye}\alpha \right)=K_{dye}\alpha .$$

If we consider the two values of fluorescence level $F_{0}$ and $F_{1}$ for unbound and bound dyes, respectively, the measured total fluorescence is $F_{tot}=c_{dye}\left(F_{0}+\left(F_{1}-F_{0}\right)\alpha \right)$. Thus, the normalized fluorescence $\Delta F\left(t\right)=\left(F\left(t\right)-F_{0}c_{dye}\right)/\left(F_{1}-F_{0}\right)c_{dye}=\alpha$ is the fraction of bound dyes. From the law of mass action, two regimes may be described for α. At short times, when $l_{Hp}\left(t\right)<l_{0}c_{dye}$, the concentration of bound dyes follows $\left(c_{dye}+K_{dye}\right)\alpha ≃l_{Hp}\left(t\right)/l_{0}$. Then, the fraction of bound dyes saturates: $\alpha ≃1-K_{dye}l_{0}/l_{Hp}\left(t\right)$. Thus, during the first regime, the fluorescence is increasing linearly with the length $l_{Hp}\left(t\right)$ of the hairpins ($F_{tot}≃F_{0}c_{dye}+\left(F_{1}-F_{0}\right)l_{Hp}\left(t\right)/l_{0}\left(c_{dye}+K_{dye}\right)$). Then comes the second regime where saturation occurs: $F_{tot}≃F_{1}c_{dye}-\left(F_{1}-F_{0}\right)K_{dye}c_{dye}l_{0}/l_{Hp}\left(t\right)$.

In general, the LAMP protocols use a fluorescent dye such as EvaGreen or SYBR Green. Depending on the choice of the dye, the parameters $l_{0}$ and the affinity constant $K_{dye}$ may differ. In the following experiments to validate the model, we considered the EvaGreen dye at a concentration 1X or 0.25X corresponding, respectively, to 1.25μM and 0.31μM. Thus, its dissociation constant is higher than the dye concentration: $K_{dye}≃3\;\mu$M [56]. We consider that $c_{dye}<K_{dye}$. So, we may assume that the dsDNA strands are not saturated by dyes at any times: $c_{dye}\alpha \ll l_{Hp}\left(t\right)/l_{0}$. In this case, the normalized fluorescence follows a logistic function:(16)$$\Delta F\left(t\right)=\alpha =\frac{l_{Hp}\left(t\right)/l_{0}}{l_{Hp}\left(t\right)/l_{0}+K_{dye}}=\frac{1}{1+\left[\left(K_{dye}l_{0}\right)/\left(l_{D}c_{D}\right)\right]\exp\left(-2k_{hyb}c_{P}t\right)}$$
where we replaced the equation of the length of hairpins $l_{Hp}\left(t\right)≃l_{D}c_{D}\exp\left(2k_{hyb}c_{P}t\right)$ in the primer-limited regime (see the Appendix A). Interestingly, the normalized fluorescence follows a logistic function with a growth rate $k=2k_{hyb}c_{P}$ and a time at half-saturation $t_{50}=\ln\left[\left(K_{dye}l_{0}\right)/\left(l_{D}c_{D}\right)\right]/\left(2k_{hyb}c_{P}\right)$. Similarly to the concentration of hairpins, the growth rate is proportional to the concentration of primers and $t_{50}∼\ln c_{D}$. In case of the dNTP-limited regime, the expression of $l_{Hp}\left(t\right)$ should be replaced by Equation (Equation 14) with the growth rate $k=2k_{E}$ related to the enzymatic rate but independent of $c_{P}$.

## 4. Experimental Results and Discussion

In order to support this theoretical model, several experiments have been performed starting from a small number of dumbbells and varying different parameters such as the concentrations of fluorescent dyes, dumbbells, primers and dNTPs. The aim of those experiments was to assess some of the predictions obtained from the kinetic models developed for IDEA.

### 4.1. Effect of Fluorescent Dye Concentrations

The exponential amplification of dumbbells is obtained by putting a solution containing the primers, FIP and BIP, a specific enzyme (Bst 2.0 DNA polymerase), dNTPs, the fluorescent dye EvaGreen and the dumbbells at 65 ${}^{\circ }$C during 1 h. First, we performed the amplification monitoring with two concentrations of EvaGreen intercalating dye (1X and 0.25X) in order to check that the saturation of the fluorescence signal was set by the dye concentrations (see Appendix A). In both cases, the ratio between the signal obtained at 60 min and the initial signal is close to 4. It corresponds to the ratio of dye concentrations present for each of the dumbbell concentrations considered (from 10 nM to 10 pM). Such experiment is interesting to confirm that the saturation of the signal is mainly set by the intercalated dyes. However, it is important to stress that the saturation of the fluorescent level observed is not indicative of the saturation of the hairpin length. The limited number of dyes is just masking the amplification reaction of the hairpin length. Thus, it prevents from differentiating the saturation due to primers or due to dNTPs concentrations. In order to experimentally observe this saturation, a different technique should be considered such as the DARQ method for example (see Section 3.4) [52]. Nonetheless, we have shown that the dependence of the exponential growth rate *k* upon the various parameters may differ between both models. We will analyze such dependence in the following sections in order to validate the models.

### 4.2. Effect of Primer and dNTP Concentrations

The primer-limited model predicts that the rate *k* is proportional to the primer concentration. Thus, by varying this concentration, we may expect to assess the prediction. We performed the dumbbell amplification for three primer concentrations ($c_{P}=3.2,4.8$ and 7.2μM, respectively) as well as for three dumbbell concentrations ($c_{D}=1000,100$ and 10 pM, respectively). As observed on Appendix A, the exponential rates are primer concentration dependent. However, by scaling the time proportionally to the primer concentration: $c_{P}\left(t-t_{0}\right)$, it may be seen (Figure 3) that the experimental signals observed for all primer concentrations overlap each other at least for the dumbbell concentrations $c_{D}=1$ nM and 100 pM (in blue and orange). This is a confirmation of the primer-limited model prediction that the growth rate is proportional to $c_{P}$: $k≃c_{P}k_{hyb}$.

A slight resetting of the initial time by $t_{0}=4$ min was necessary to obtain strong overlap of experimental results. This resetting time may be explained by the timescale required to transform the dumbbells in Hp1b hairpins which was neglected in the model. It could also be due to an experimental issue to correctly determine the initial time of the amplification regime. Thus, the time $t_{0}$ could also reflect the time needed for the solution to reach the required temperature for the enzymatic amplification.

For the lower dumbbell concentration of $c_{D}=10$ pM, the amplification signals for the different primer concentrations are clearly different even with the time scaling suggesting that the dNTP concentration may saturate in this case. Indeed, this discrepancy was also slightly observable at later times (or near saturation) for the dumbbell concentration $c_{D}=100$ pM. The experimental results are compatible with the amplification model described previously that switches from a primer-limited to a dNTP-limited regime at the crossover time $t_{co}$ (Section 3.3).

Those results are also confirmed by the amplification performed for three different dNTP concentrations (0.93,1.4 and 2.1 mM, respectively) where no dependence on dNTP concentration is observed (see Appendix A) for dumbbell concentrations $c_{D}=1$ nM and 100 pM while a dependence on dNTP slightly occurs for $c_{D}=10$ pM. In order to analyze in more details the influence of dNTP concentrations, a larger range of concentrations on several order of magnitudes should be considered as well as another experimental technique in order to assess the saturation due to the lack of dNTPs.

### 4.3. Effect of Dumbbell Concentration and Length

Since we have experimentally confirmed the dependence of the exponential growth rate $k∼c_{P}$ to the primer concentration (at least for short times before the crossover $t_{co}$), we are now addressing the dependence of the time $t_{50}$ when the signal reaches half of the saturation. In order to confirm the prediction about $t_{50}$, the amplification reaction of a range of six orders of magnitude in dumbbell concentrations (from $c_{D}=1$ nM to 1 fM) was performed for all the dumbbells with varying lengths ($l_{D}=134,153$ and 190 for SD, MD and LD, respectively). Those results are presented in Figure 4 for the dumbbell MD with the logistic fit (black lines) with two parameters: the growth rate *k* and the time $t_{50}$ at half saturation. As can be seen on the figure, despite the approximations used in the theoretical models, the fits match fairly well the experimental data.

Interestingly, for most of the concentrations, the fitted values of $t_{50}$ differ less than 5% from the experimental values defined as the time to reach 50% of the signal saturation. The results show that this time $t_{50}$ is a function of the dumbbell concentration for the three dumbbells considered in Figure 5 and confirm that $t_{50}$ linearly scales with $\ln c_{D}$. Furthermore, we observe that there is only a slight difference for the three dumbbell lengths suggesting that $t_{50}$ is mostly independent of this parameter as expected with the model in the primer-limited regime. While the amplification is slightly faster for the short dumbbell SD, the two larger dumbbells MD and LD are amplified at the same rate.

Finally, we aim to illustrate the predictive opportunity brought by our theoretical model. We have performed the amplification reaction over six orders of magnitude in the dumbbell concentrations. Let us consider that one concentration, $c_{D}\left[Ref\right]=1$ nM, is the reference. For this reference, we are able to determine the time at half saturation $t_{50}\left[Ref\right]$ and the growth rate *k* by fitting the experimental data with a logistic function as illustrated in Figure 4. From those fitted values, it is then possible to predict the time $t_{50}\left(c_{D}\right)$ for all concentrations $c_{D}$:(17)$$t_{50}\left(c_{D}\right)=t_{50}\left[Ref\right]-k^{-1}\ln\left(c_{D}/c_{D}\left[Ref\right]\right).$$
In Table 2, we presented the theoretical predictions in comparison with the experimental data for the three dumbbell lengths considered and six orders of magnitude in concentrations. For all the concentrations and dumbbell lengths, the theoretical predictions are in good agreement with the experimental data with an error less than 10% for all cases and generally within only few percent differences.

## 5. Conclusions and Outlook

In this study, we have set the theoretical framework for the understanding of the exponential amplification kinetics of LAMP experiments as function of the physico-chemical parameters of the reaction. We particularly emphasized on the concentrations of initial strands to amplify (the dumbbells) but also on the concentrations of the primers and dNTPs used for the amplification reaction. Our IDEA kinetic model allows for explicit expressions of the kinetics evolution of the concentration of hairpins (or amplicons) as well as their total length. In the simplest case of primer-limited regime, the hairpin concentration kinetics follows a logistic function. However, in general LAMP experiments, the measured signal is the fluorescence of intercalating dyes inside the double stranded stems of the hairpins. This signal has been shown to be related to the length of the hairpins formed which increases quadratically with the concentration of hairpins. Some of our predictions were experimentally confirmed: we have varied the concentrations of various parameters such as the concentrations of dumbbells, primers, dNTPs and fluorescent dyes as well as the length of the dumbbell. The experimental results are in agreement with an exponential growth rate proportional to the primer concentration in the primer-limited regime. This regime then switches to a dNTP-limited regime with a growth rate related to the enzyme. The half-saturation time $t_{50}$ scaled logarithmically with the dumbbell concentration independently of its length.

Our model was developed for the amplification of dumbbell probes, the second stage of LAMP. However, we believe that it could be generalized to standard LAMP amplification reactions by considering also the first stage in the model. We are also convinced that the general framework could be used for various alternatives of LAMP that recently appeared in the literature for the detection of targets. In the case of aptamero-LAMP for protein detection [37,38], the generalization is straightforward since the first stage consists in selecting dumbbell probes forming a sandwich with the protein target to be detected. In the detection of miRNA [24], the first stage of LAMP consists in forming a dumbbell by the simple ligation of two hairpins strands. Obviously, our model was based on several assumptions and approximations. Thus, further theoretical developments could be useful to improve the predictions. Furthermore, new LAMP functionalities could be incorporated into the model. For example, LAMP rate may be increased by the use of loop primers [57]. It would be interesting within this framework to understand the reason of the LAMP acceleration due to the loop primers. Finally, we have discussed that several experimental techniques are measuring either the concentration (DARQ method) or the length of hairpins (intercalating dyes or gel electrophoresis) of the LAMP reaction. It would be interesting to combine those techniques to characterize the amplification more deeply.

## Figures and Tables

**Figure 1 biosensors-12-00346-f001:**
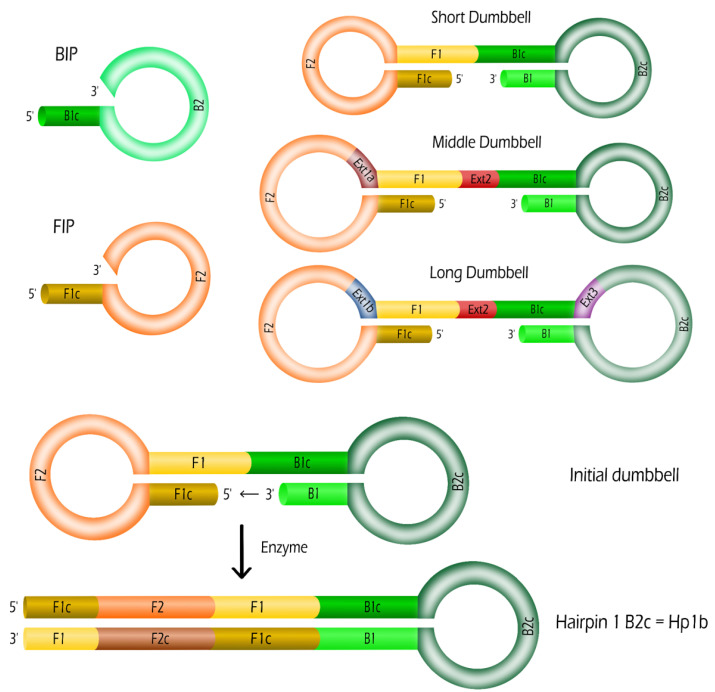
The primers (BIP and FIP) and the dumbbells (Short dumbbell SD, middle dumbbell MD and long dumbbell LD) are depicted with their name sequences and respective extensions. At the bottom, the formation of the hairpin Hp1b from SD is obtained from an enzymatic extension of the 3${}^{\prime }$ end.

**Figure 2 biosensors-12-00346-f002:**
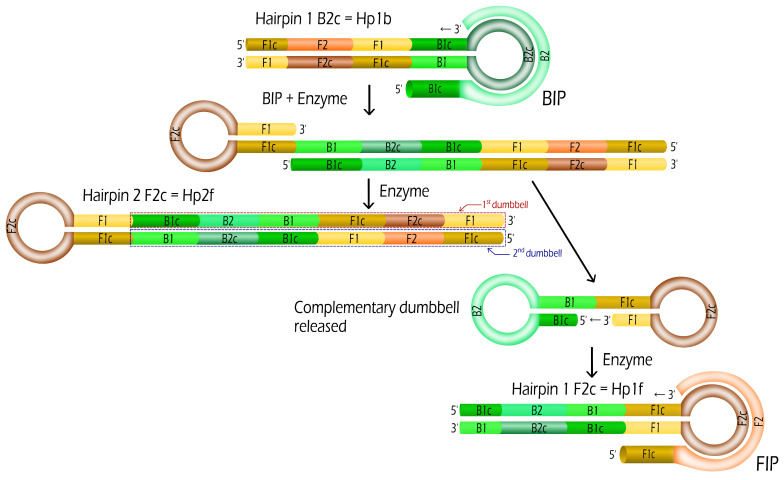
Duplication: Two hairpins Hp1f and Hp2f are created from the hairpin Hp1b with the use of a single BIP and enzymes.

**Figure 3 biosensors-12-00346-f003:**
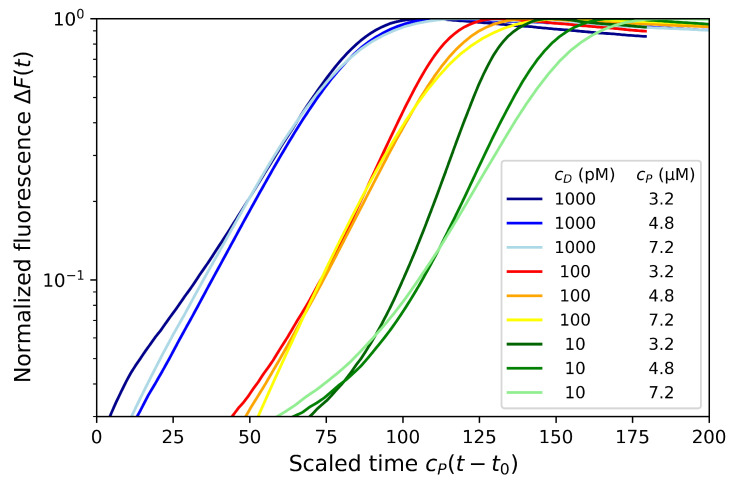
Normalized fluorescence ΔF in logarithmic scale on the y-axis as function of scaled time $c_{P}\left(t-t_{0}\right)$ (in μM.min) for three concentration of primers ($c_{P}=3.2,4.8,7.2$μM) and three MD dumbbell concentrations $c_{D}=1000,100,10$ pM.

**Figure 4 biosensors-12-00346-f004:**
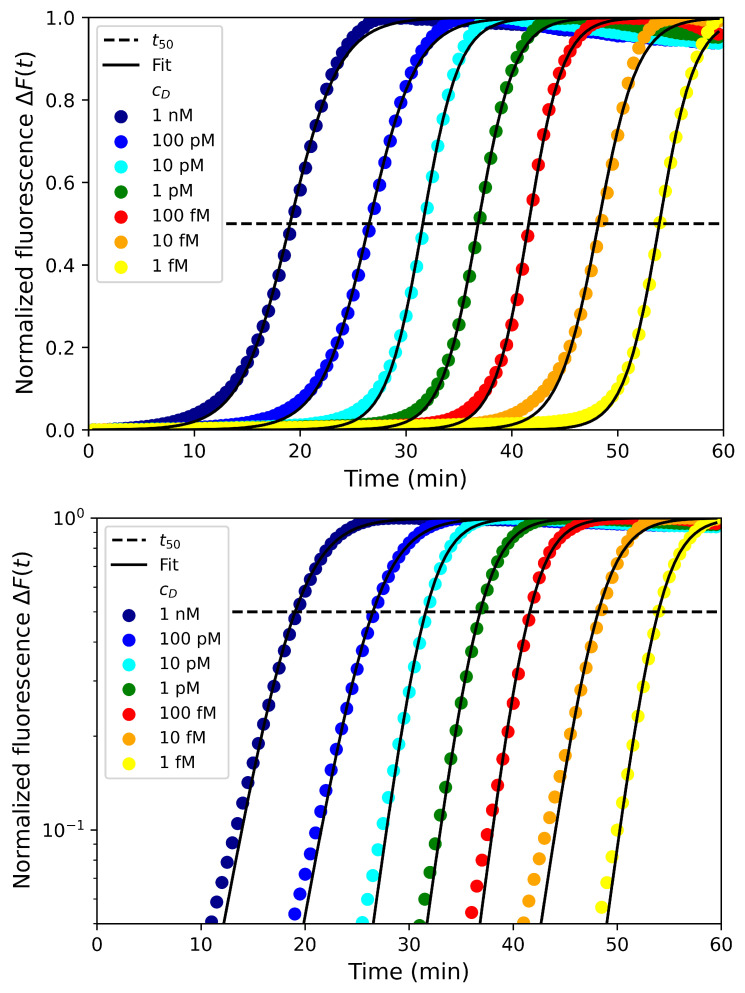
Normalized fluorescence ΔF in linear (**above**) and logarithmic (**below**) scales as function of time for various concentrations of the dumbbell MD from 1 nM to 1 fM. The black dashed line is the half-saturation threshold for the determination of $t_{50}$. The dark lines are the logistic fit of the experimental data.

**Figure 5 biosensors-12-00346-f005:**
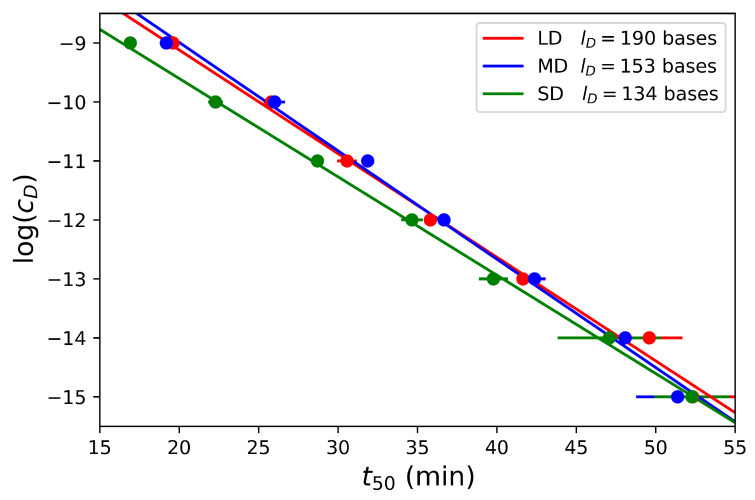
Dumbbell concentrations in logarithmic scale as function of time $t_{50}$ for various concentrations of dumbbells from 1 nM to 1 fM and for the three different dumbbell lengths.

**Table 1 biosensors-12-00346-t001:** Oligonucleotide sequences and dumbbell structures.

Name	Length	Sequence (5${}^{\prime }$-3${}^{\prime }$)
B1	25	GGG GGA AAG ATA TAA CTC AGA GAT G
B2	18	GAA GGA GGG TCA GTG AGG
F1	21	ATA AAC CGC GTC TTG GAT CCG
F2	24	CGT GCA GTA CGC CAA CCT TTC TCA
FIP	45	F1c-F2
BIP	43	B1c-B2
Ext1a	16	TGC GCT GCC CCT CTT A
Ext1b	23	TGC GCT GCC CCT CTT ATA TCT TC
Ext2	3	TGA
Ext3	30	AGT TTA CAG CTC CTT AAG CCC CAT ATT GCC
SD	134	F1c-F2-F1-B1c-B2c-B1
MD	153	F1c-F2-Ext1a-F1-Ext2-B1c-B2c-B1
LD	190	F1c-F2-Ext1b-F1-Ext2-B1c-Ext3-B2c-B1

**Table 2 biosensors-12-00346-t002:** Experimental (*n* = 2) versus theoretical predictions for the time at half saturation $t_{50}$ in minutes as function of the dumbbell lengths and concentrations. The experimental error is set as the standard deviation values calculated from the replicates. The fit at the reference concentration (1 nM) was used to determine $t_{5}0$[Ref] and *k*. The theoretical values were then determined from Equation (Equation 17). The column named Error reflects the variation between the experimental results (values at half saturation) and the theoretical predictions.

Dumbbell	SD = 134 Bases			MD = 153 Bases			LD = 190 Bases		
$c_{D}$	**Exp.**	**Theo.**	**Error**	**Exp.**	**Theo.**	**Error**	**Exp.**	**Theo.**	**Error**
1 nM	16.9±0.1	16.6	1.8%	19.2±0.1	19.0	1.0%	19.6±0.5	19.4	1.2%
100 pM	22.3±0.5	22.3	0.1%	26.0±0.7	24.3	6.5%	25.8±0.5	24.6	4.6%
10 pM	28.7±0.1	27.9	2.8%	31.9±0.1	29.7	6.9%	30.6±0.6	29.9	2.3%
1 pM	34.6±0.7	33.6	2.9%	36.7±0.2	35.0	4.6%	35.8±0.1	35.1	2.0%
100 fM	39.8±0.9	39.2	1.5%	42.4±0.7	40.3	5.0%	41.6±0.1	40.3	3.1%
10 fM	47.1±3.2	44.8	4.9%	48.1±0.4	45.7	5.0%	49.6±2.1	45.6	8.1%
1 fM	52.3±2.4	50.5	3.4%	51.7±2.6	51.0	1.4%	52.3±2.8	50.8	2.9%

## Data Availability

Data are available upon request.

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
