# Peer review of "Kinetics of Isothermal Dumbbell Exponential Amplification: Effects of Mix Composition on LAMP and Its Derivatives"

_biosensors, 2022, doi:10.3390/bios12050346_

Round 1
Reviewer 1 Report
Authors breakdown the well know isothermal amplification method, LAMP, into two stages, i.e., the first one consists in the transformation of the DNA strands into dumbbell structures formed of two stems and loops thanks to four primers, then, in the second stage, only two primers are required to amplify the dumbbells exponentially in numerous hairpins of increasing lengths. Authors propose a theoretical framework to analyze the kinetics of the second stage of LAMP process, which is called, the isothermal dumbbell exponential amplification (IDEA) as logistic function of the physico-chemical parameters, principally the concentration of dumbbells, primers and nucleotides, to better analyze the experimental signal growth based on primer hybridization limiting reaction rate and concentrations. Authors are able to predict the increase of the amplicons (or hairpins) concentration with the dependence of k and t50 on the concentration of dumbbell and primers and with experimental validations of predictions.
The manuscript is prepared in clear and concise format. The theoretical framework of IDEA amplification part can well descript the amplicons behavior with various limiting conditions, I think this is a novel idea to me.
Authors can help to explain how this framework can be used to better descript the unique smear gel pattern of LAMP amplification process, which has various length of amplicons in the theoretical treatment, or comment on this outcome.
Reviewer 2 Report
See Word file

Reviewer 3 Report
Journal Biosensors (ISSN 2079-6374)
Manuscript ID biosensors-1686967
Title Kinetics of Isothermal Dumbbell Exponential Amplification: Effects of Mix Composition on LAMP and its Derivatives
Authors Maud Savonnet , Mathilde Aubret , Patricia Laurent , Yoann Roupioz , Myriam Cubizolles * , Arnaud Buhot *
Section Biosensor and Bioelectronic Devices
Special Issue Advances in Amplification Methods for Biosensors
This manuscript describes the set the theoretical framework for the understanding of the exponential amplification kinetics of LAMP experiments as function of the physico-chemical parameters of the reaction.
The manuscript is clearly described, well motivated and pertinent to the journal.
The plan structure and tasks are clearly presented.
Also, I find the described experiments to be balanced and the relevant literature has been cited.
I find this manuscript a well written, clear and an interesting, timely report. So my recommendation would be to Accept in present form.
